# SEMANTIC FEATURE VERIFICATION IN FLAN-T5

**Siddharth Suresh, Kushin Mukherjee & Timothy T Rogers**
Department of Psychology
University of Wisconsin-Madison
Madison, WI 53706, USA
{siddharth.suresh, kmukherjee2, ttrogers}@wisc.edu

## ABSTRACT

This study evaluates the potential of a large language model for aiding in generation of semantic feature norms –a critical tool for evaluating conceptual structure in cognitive science. Building from an existing human-generated dataset, we show that machine-verified norms capture aspects of conceptual structure beyond what is expressed in human norms alone, and better explain human judgments of semantic similarity amongst items that are distally related. The results suggest that LLMs can greatly enhance traditional methods of semantic feature norm verification, with implications for our understanding of conceptual representation in humans and machines.

**Introduction.** In cognitive science, efforts to understand the structure of human concepts have relied on semantic feature norms: participants list all the properties they believe to be true of a given concept; responses are collected from many participants for many concepts; overlap in the resulting feature vectors captures the degree to which concepts are semantically related(Rosch, 1973; McRae et al., 2005). Yet participants often produce only a fraction of what they know for each concept: tigers have DNA, can breathe, and are alive, but these properties are not typically produced in feature norms for *tiger*. Such omissions are important because they express deep conceptual structure: having DNA and breathing connect tigers to all other plants and animals. To better capture such structure, some studies ask human participants to make yes/no judgments for *all possible properties* across every concept. Thus if "can breathe" was listed for a single concept, human raters would then evaluate whether each other concept in the dataset can breathe. This *verification* step significantly enriches the conceptual structure that features norms express (De Deyne et al., 2008), but is exceedingly costly in human labor: the number of verification questions asked increases exponentially with the number of concepts probed. Previous work has shown that the conceptual structure of a large language model (LLM) for semantic feature listing is similar to human conceptual structure (Suresh et al., 2023; Bhatia & Richie, 2022). In this paper we consider whether this step can be reliably "outsourced" to an open sourced LLM optimized for question-answering, specifically the open-source FLAN-T5 XXL model (Chung et al., 2022; Wei et al., 2021), focusing on two questions: (1) How accurately does the LLM capture human responses to the questions? (2) Do the LLM-verified feature vectors better capture human-perceived semantic structure amongst the concepts?

**Method.** We used all animal (129) and artifact (166) names from the Leuven semantic norms (De Deyne et al., 2008). In the generation phase of that study, 1003 participants were asked to list 10 semantic features for 6-10 different words. Features were agglomerated across all items to produce a 2600-dimensional feature vector. In a subsequent verification phase, four raters evaluated concept-feature pairs within the animals and within the artifacts, judging whether each feature was true of each concept. Importantly, raters did not evaluate cross-domain features, such as whether a tiger "has wheels" or whether a pillow "has fur." Thus the norms express very little similarity between animals and artifacts. To outsource the verification phase, we probed FLAN-XXL with yes/no queries asking, for each concept, whether it possesses each feature, including both within-domain questions (like the original study) and between-domain questions. The model responded to 597,670 probes, yielding a *machine-verified* binary feature matrix in which every cell where the model affirmed that concept $C$ had feature $F$ was filled with a 1 and remaining cells were filled with 0s.

**Results.** From human norms each matrix cell was classified as a *target* if all raters agreed the corresponding concept had the corresponding property and a *distractor* otherwise. Each cell in the

machine-verified matrix was then classified as a hit, miss, false-alarm, or correct rejection relative to the human matrix. From these data we computed the hit and false-alarm rates for the machine-verified matrix and converted these to the $d'$ measure of signal separability (Stanislaw & Todorov, 1999). For the whole matrix, $d'$ was 2.11 (hr = 0.76, far = 0.1); considering animals only $d'$ was 1.48 (hr=0.70, far=0.19); considering artifacts only, $d'$ was 2.10 (hr=0.77, far=0.11). Thus while FLAN-XXL responses align moderately well with the human data, the machine misses 20-30% of properties verified by humans, and asserts many properties not verified in the human data.

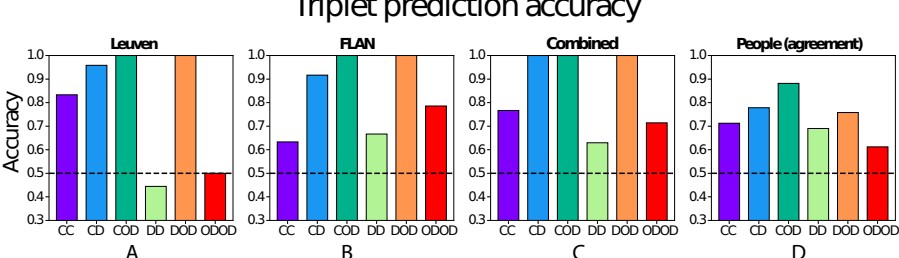

Figure 1: Given a sample item from a particular category (among the 13 Leuven categories) and domain (animal, artifacts), the two option items could be (1) both from the same category as the sample (CC condition), (2) one from the same category and one from a different category in the same domain (CD), (3) one from the same category and one from a different domain all together (CO), (4) both from a different category in the same domain (DD), (5) one from a different category in the same domain and one from a different domain (DOD), or (6) both from a different domain (ODOD).(A) Representations obtained from just the leuven norms (B)Representations obtained from FLAN (C) Representations obtained from combining Leuven norms along with FLAN responses of cross-domain features (D) Percentage of people who agreed with the majority vote.

Many of the "missed" properties clearly represent failures of the AI; for instance, FLAN-XXL denies that a horse has eyes and that a rooster lays eggs. Yet many of the false-alarms, by inspection, may reflect shortcomings of the human data. For instance, the human norms deny that a dog "can become dirty" or that a car is "not eaten," where FLAN-XXL asserts the opposite. Moreover, FLAN-XXL asserts many properties in common across animal and artifact domains that were not included in the verification phase of the original study–for instance, "constitutes a whole" or "comes in different kinds." Thus it is unclear whether the discrepancies between original and machine-verified feature sets primarily reflect shortcomings of the AI or of the human verification procedure.

To evaluate these possibilities, we computed cosine distances between all concepts in each matrix, taking these as a proxy for semantic similarity structure in the human mind (McRae et al., 2005). From each matrix we predicted human decisions in a triplet comparison task, in which participants must decide which of two option words is most similar in meaning to a target word A.5. Triplets were designed so that each option word could be drawn either from the same category as the target word (C), from a different category in the same domain (D), or from the other domain (OD). For each triplet, we computed the predicted response from the human-verified or from the machine-verified semantic distance matrices, then computed, for each triplet type, how often the model prediction agreed with the human majority-vote. Results are shown in Figure 1. Human- and machine-verified spaces did equally well when one option was semantically much closer than the other (CD, COD, and DOD conditions). When both options were semantically close to the target, the human-verified space outperformed the machine-verified space; however, the reverse was true when the two options were both distal to the target (DD and ODOD conditions). Distance matrices derived from combined human- and machine-features achieved the best of both worlds, expressing local semantic similarities as well as the human-only space and distal similarities as well as the machine-only space.

**Conclusion.** While large question-answering models like FLAN-XXL cannot completely supplant human effort in verifying semantic feature norms, they capture aspects of conceptual structure beyond that expressed in human norms alone. Combining human- and machine-verified data may provide the most accurate estimates of human semantic structure, which in turn represents a critical step for understanding conceptual representation in both natural and artificial intelligence. We can also get more accurate feature norm estimates by combining human- and machine-verified data as shown by Mukherjee et al. (2023).

URM STATEMENT

"The authors acknowledge that at least one key author of this work meets the URM criteria of ICLR 2023 Tiny Papers Track."

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

## A  APPENDIX

You may include other additional sections here. However, please be mindful that the spirit of the Tiny Papers track is for papers to be short. Avoid overly-long appendices.

### A.1  $d'$ AS AN ALIGNMENT METRIC

Since both the ground-truth human feature matrices and LLM-generated feature matrices had binary entries, the problem of human-machine comparison can be described in terms of signal detection theory. That is, if we treat the human-matrix as being the source of 'signal', the predictions of 1s and 0s in the machine matrix can be — (1) Hits if the cell in the matrix was 1 for both the human and machine matrix, (2) Misses if the cell in the machine matrix is 0 and 1 in the human matrix, (3) False alarms if the cell in the machine matrix is 1 and 0 in the human matrix, and (4) Correct rejections if the cell in both the machine and human matrices is 0. The number of hits, misses, false alarms, and correct rejections can be tallied to compute hit rate (HR) and false-alarm rate (FAR) as follows -

$$HR = \frac{hits}{hits + misses} \tag{1}$$

$$FAR = \frac{false\ alarms}{false\ alarms + correct\ rejections} \tag{2}$$

Finally, $d'$ is computed as

$$d' = z(H) - z(FAR), \tag{3}$$

where $z$ is the inverse of the cumulative distribution function (CDF) of the standard normal distribution $\mathcal{N}(0,1)$. A higher $d'$ indicates a greater degree of alignment between human and machine features.

## A.2 FLAN-T5 FEATURE VERIFICATION

We queried FLAN-T5 XXL to assess whether a property was associated with a concept. For example, to asses if the model believed that dolphins have two eyes, we probed it with the prompt shown in the table below.

Table 1: Prompt used while querying FLAN

Q: Is the property [is_female] true for the concept [book]?
A: False
Q: Is the property [can_be_digital] true for the concept [book]
A: True
In one word True/False, answer the following question
Q: Is the property [has_two_eyes] true for Dolphins?
A: <mask>

## A.3 SENSITIVITY ANALYSIS OF FLAN AT DIFFERENT LEVELS OF INTER-RATER AGREEMENT

During the feature verification phase of the Leuven study De Deyne et al. (2008), four raters were employed to assess the association between a feature and concept. We investigated the impact of threshold levels on d' values by applying various thresholds to the Leuven feature matrix. A threshold of 25% was used to indicate that a concept-feature association was valid if only one of the four raters agreed, whereas a threshold of 100% was used to indicate that all four raters agreed on the validity of a concept-feature association.

Table 2: Sensitivity analysis of Flan-XXL in the animal domain. d' increases as we increase the inter-rater agreement.

| Inter-rater agreement | d' | hit-rate | false-alarm rate |
|---|---|---|---|
| 25% | 1.11 | 0.48 | 0.13 |
| 50% | 1.28 | 0.58 | 0.15 |
| 75% | 1.43 | 0.67 | 0.17 |
| 100% | **1.48** | 0.70 | 0.19 |

Table 3: Sensitivity analysis of Flan-XXL in the artifacts domain. d' increases as we increase the inter-rater agreement.

| Inter-rater agreement | d' | hit-rate | false-alarm rate |
|---|---|---|---|
| 25% | 1.66 | 0.51 | 0.05 |
| 50% | 1.85 | 0.64 | 0.07 |
| 75% | 2.06 | 0.75 | 0.09 |
| 100% | **2.20** | 0.81 | 0.11 |

## A.4    DIFFERENCES IN FLAN AND HUMAN FEATURE VERIFICATION

Table 4: Top 20 Features in **Leuven Animals** where FLAN and Leuven norms differ the most

| Leuven norms say yes but flan says no | Flan says yes but Leuven norms say no |
|---|---|
| lives_in_Europe | occasionally_occurs_in_films |
| is_found_in_Belgium | is_smaller_than_a_lorry |
| has_two_eyes | is_sometimes_eaten_by_man |
| lays_eggs | exists_in_different_sizes_and_kinds |
| lives_in_distant_countries | exists_in_different_sizes |
| has_eyes | can_become_dirty |
| is_smooth | can_be_brown,_black,_white,_grey |
| lives_in_the_zoo | small_and_large_kinds |
| is_not_a_pet | could_be_an_animal |
| has_four_paws | has_a_skin |
| is_light | can_be_caught |
| also_lives_in_the_city | has_a_mouth |
| is_slippery | attaches_to_the_body |
| has_six_paws | has_been_existing_for_a_long_time |
| has_legs.1 | doesn't_have_1000_paws |
| is_not_poisonous | consists_of_different_parts |
| is_an_animal | exists_in_different_forms |
| can't_fly | exists_in_different_types |
| doesn't_make_a_sound | exists_in_different_kinds |
| lives_in_the_open_air | constitutes_a_whole |

Table 5: Top 20 Features in **Leuven Artifacts** where FLAN and Leuven norms differ the most

| Leuven norms say yes but flan says no | Flan says yes but Leuven norms say no |
|---|---|
| is_hard | needs_to_be_cleaned_sometimes |
| sold_in_Gamma_(particular_hardware_store) | exists_in_different_materials |
| is_sold_in_Brico_(particular_hardware_store) | neutral_scent |
| it_is_absorbent | can_have_a_container_ |
| has_a_metallic_color | is_not_eaten |
| has_no_roof | does_not_lay_eggs |
| is_silver-coloured | used_in_different_cultures |
| played_by_a_single_person | often_used_ |
| the_size_is_indicated_by_numbers | smaller_than_a_horse |
| has_windows | small_and_large_kinds |
| hasn't_been_existing_for_such_a_long_time | there_are_many_kinds_of_it |
| is_polluting | is_found_all_around_the_world |
| is_stainless | consists_of_different_parts |
| is_loose | is_not_poisonous |
| is_easy_to_play | is_not_a_pet |
| bought_in_a_store | is_smaller_than_a_lorry |
| weighs_much | constitutes_a_whole |
| can_be_seen_on_television | was_used_in_the_past |
| costs_a_lot_of_money | exists_in_different_sizes_and_kinds |
| lies_in_a_garden_house | doesn't_have_1000_paws |

A.5   TRIPLET JUDGEMENT TASK

In the triplet judgement task, participants were presented with a reference word and two option words, and had to decide which option was more similar in meaning to the reference word.

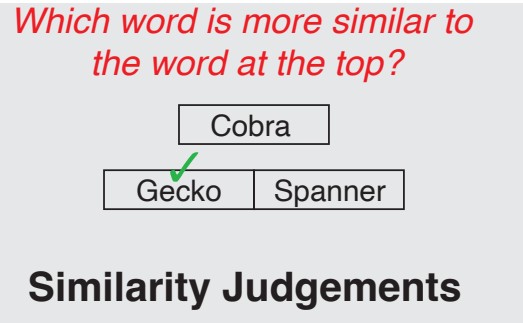

Figure 2: Triplet judgment task example

