# OpenReview forum: "Semantic feature verification in FLAN-T5"
_ICLR.cc/2023/TinyPapers — Submitted to Tiny Papers @ ICLR 2023_

### Official Review · Reviewer_P5d7 · 2023-03-30

**Confidence:** 5

**Summary Of Contributions:**

This paper study the potential use of LLMs for generating semantic feature norms, These norms are critical while evaluating concepts in cognitive science. This study shows that though LLMs missed to predict many obvious properties but overall help improve human understanding of different concepts and helps better explaining human judgments of semantic similarity among items that are distally related.

**Rating:**

Clear, Correct, and Reproducible (CCR): a submission which meets the reviewing criteria

**Strengths And Weaknesses:**

- Clarity: Paper is clearly written and relevant literature is discussed.

- Correctness: Claims and conclusions are more or less justified by the findings. I say more or less because LLMs still missed to predict several properties which are obvious ones and hence makes me doubt how the use of LLms helps humans explaining semantic similarity among items that are distally related.

- Reproducibility: No code and data is provided

- Follows basic requirements: yes

**Suggested Changes:**

- Comments on presentations
	- use `' as quotations marks.

---

### Official Review · Reviewer_49HX · 2023-04-01

**Confidence:** 4

**Summary Of Contributions:**

This paper presents an evaluation scheme showing how large language models can be used for semantic feature verification.

**Rating:**

Needs Clarification (NC): a submission which does not meet the reviewing criteria and needs clarification for its described problem or solution

**Strengths And Weaknesses:**

### Strength ###
- Follows basic requirements: submission adheres to formatting requirements and the page limit
- Reproducibility: paper describes its methods in such detail that a reader could independently reproduce the findings

### Weaknesses ###
- Clarity: Although the authors present a case where LLMs can be used to add new features with better alignment to local and global semantic relations in human cognition, from the results presented in Figure 1 it is not clear whether they support this conclusion.

**Suggested Changes:**

- Can the authors provide numerical values of the results shown in Figure 1?
- There might a typographical error on page 2, in the paragraph before the conclusion. The sentence says 'Results are shown in Figure 2', but I believe the authors are referring to Figure 1. Please clarify.

---

### Meta-Review · Area_Chair_g2Ty · 2023-04-07

**Recommendation:** Invite to present
**Confidence:** 5

**Metareview:**

The paper is clear, and original and clearly motivates the use of LLMs for semantic feature norms. This is one of the areas where Instruction-tuned LLMs models can be used to further enhance human understanding of specific concepts.

**Summary:**

This works mainly studies the use of LLMs for generating semantic feature norms. The authors did a good job motivating the use of LLms for this task and clearly lay down the experimental setting. One of the major concerns is that the main claims of the work are not fully justifiable from the results presented.

**Reason For Not Giving A Higher Recommendation:**

NA

**Reason For Not Giving A Lower Recommendation:**

Though the claims are not fully justifiable, I believe this work shows the potential use of LLMs in semantic feature norms. Readers may get benefited from the presented analysis and approach.

---

### Decision · Program_Chairs · 2023-04-08

Invite to present